# Beyond Loading: Functions of Plant ARGONAUTE Proteins

**DOI:** 10.3390/ijms242216054

**Published:** 2023-11-07

**Authors:** Chao Liang, Xiaoliu Wang, Hualong He, Chi Xu, Jie Cui

**Affiliations:** Guangdong Provincial Key Laboratory for Plant Epigenetics, College of Life Sciences and Oceanography, Shenzhen University, Shenzhen 518060, China; chaoliang@szu.edu.cn (C.L.); 2200251012@email.szu.edu.cn (X.W.); 2020305019@email.szu.edu.cn (H.H.); xchisz@szu.edu.cn (C.X.)

**Keywords:** ARGONAUTE proteins, small-RNA, loading, non-canonical functions

## Abstract

ARGONAUTE (AGO) proteins are key components of the RNA-induced silencing complex (RISC) that mediates gene silencing in eukaryotes. Small-RNA (sRNA) cargoes are selectively loaded into different members of the AGO protein family and then target complementary sequences to in-duce transcriptional repression, mRNA cleavage, or translation inhibition. Previous reviews have mainly focused on the traditional roles of AGOs in specific biological processes or on the molecular mechanisms of sRNA sorting. In this review, we summarize the biological significance of canonical sRNA loading, including the balance among distinct sRNA pathways, cross-regulation of different RISC activities during plant development and defense, and, especially, the emerging roles of AGOs in sRNA movement. We also discuss recent advances in novel non-canonical functions of plant AGOs. Perspectives for future functional studies of this evolutionarily conserved eukaryotic protein family will facilitate a more comprehensive understanding of the multi-faceted AGO proteins.

## 1. Introduction

ARGONAUTE (AGO) proteins are a conserved family of eukaryotic proteins that act as core components of RNA silencing by loading small RNAs (sRNAs) derived from plants and other interacting organisms [1]. sRNAs associate with AGOs to form RNA-induced silencing complexes (RISCs), which target complementary RNA or DNA sequences to repress gene expression by post-transcriptional gene silencing (PTGS) or transcriptional gene silencing (TGS), respectively [2].

Phylogenetic analyses have revealed three major clades of angiosperm AGO proteins, which are named after *Arabidopsis* AGOs: AGO1/5/10, AGO2/3/7, and AGO4/6/8/9 [3]. Monocots have also evolved a specific AGO, AGO18, which forms its own subclade closest to the AGO1/5/10 subclade [3]. AGO17, whose evolutionary position has fluctuated outside and inside the AGO1 clade [4,5,6,7], was first identified as a rice-specific AGO family member [5] and was subsequently discovered in several other Poaceae species [8]. A recent phylogenetic tree based on genomic and transcriptomic data from more plant species revealed additional divergence of eudicot AGO10 into AGO10a and AGO10b [7]. Notably, another AGO-like subclade exists in early land plants, such as hornworts, lycophytes, ferns, and gymnosperms, but has been lost specifically in angiosperms [9].

Traditional functions of AGOs are inseparable from sRNA loading, and their multifaceted roles in plant development and environmental response are determined by the types of sRNAs with which they associate. Intensive studies of miRNA–AGO loading have revealed the mechanisms by which sRNAs are sorted into various AGOs, depending on the 5′ terminal nucleotide, the sRNA length, and the sequence and structural features of the sRNA duplex and sRNA precursors [10,11,12,13]. The spatiotemporal expression pattern of the AGO and its sRNA partner is another major factor that affects sRNA sorting [14]. However, sRNA loading does not only initiate target gene silencing through an sRNA-mediated mechanism; it has broader functions in balancing distinct classes of sRNAs, modulating plant development and defense by cross-regulation within the AGO protein family, and enabling intra- and extracellular sRNA movement. In this review, we summarize the latest research progress on these emerging, nontraditional functions of AGO proteins and outline a number of important questions for future study.

## 2. Traditional and New Roles of AGO Proteins

### 2.1. Loading Competition Balances sRNA Types

Four conserved domains constitute the typical structures of eukaryotic AGOs, including the N-terminal domain and the PAZ, MID, and PIWI domains, in which the MID provides a basic pocket for sRNA loading together with the PIWI domain [1,15,16]. Commonly, sRNA duplexes are loaded onto AGO proteins with the assistance of several chaperone proteins [17,18,19]. And then, due to the thermodynamic stability and sequence features of the sRNA duplexes, one strand (passenger strand) is degraded, and the retaining strand (guide strand of miRNA or siRNA) and AGO form the functional RISC to guide silencing of targets [20]. As key components of sRNA-mediated target-gene silencing, AGO proteins maintain a balance between the two major classes of plant sRNAs: miRNAs and small interfering RNAs (siRNAs). AGO1 is the predominant component responsible for the loading of most miRNAs and some siRNAs; therefore, miRNA–siRNA competition can occur. As a result, AGO protein availability is one of the factors that limit transgene-siRNA-mediated PTGS, and efforts to increase AGO1 protein levels also enhance PTGS efficiency. In this scenario, PTGS efficiency is limited by the miRNA pathway and, thus, protects endogenous mRNAs from routine degradation [21]. The same situation can occur during plant–virus interactions: competitive binding to AGO proteins leads to the formation of two cellular AGO1 pools in vivo, one preferentially loaded with siRNAs and the other one with miRNAs. The plant PTGS pathway also generates siRNAs that are derived from invading viral RNAs. These exogenous sRNAs could load onto host AGOs and target complementary viral RNAs for degradation, leading to the suppression of virus infection [22]. However, viruses encode RNA-silencing suppressors (VSRs) that hinder AGO–sRNA loading to increase their infection ability [23], and the loading efficiencies of the two distinct cellular AGO1 pools are affected differently by VSRs [24]. AGO protein availability also represents a major determinant for the loading control of excess miRNAs, as the unbound pool of free miRNAs will be loaded into AGOs to form high-molecular-weight RISCs when AGO proteins are more abundant [25]. Nonetheless, there is a limit above which overexpression of AGO1 can further enhance the efficiency of target-gene silencing, as the supply of RISC assembly assistant factors, such as the cyclophilin 40 SQUINT (SQN) and heat shock protein 90 (HSP90), are also limited [26].

Defects in the plant RNA-decay pathway often result in the production of aberrant siRNAs to initiate the AGO1-dependent PTGS pathway, which acts to eliminate dysfunctional endogenous RNA transcripts. Examples include defects in the mRNA 5′-decapping complex and mutations in genes associated with bidirectional cytoplasmic RNA decay [27,28]. Abnormal processing of rRNA precursors (pre-rRNAs) caused by defects in exonuclease activity also leads to pre-rRNA accumulation and production of rRNA-derived siRNAs (risiRNAs) through the PTGS pathway. In the study of [29], the predominant risiRNAs were 21 nt in length, and an AGO-immunoprecipitation (AGO-IP) assay demonstrated their ability to compete with miRNAs for loading on AGO1 and AGO2, which resulted in a decline in endogenous miRNA levels [29]. Although the AGO–siRNA mediated PTGS pathway has been shown to eliminate abnormal endogenous RNAs that cannot be targeted for degradation by exonucleases, the biological role of the risiRNA–AGO complex remains to be fully clarified.

In addition to endogenous miRNAs and aberrant siRNAs, sRNAs derived from other sources also competitively bind AGO proteins to direct important developmental processes or environmental responses [30]. Loading of tRNA-derived RNA fragments (tRFs) onto AGOs was demonstrated back in 2013. Most AGO-associated tRFs were 18~22 nt in length with 19 mers predominating. The 5′ nucleotide appeared not to determine the sorting of tRFs into different AGOs, as the protein IP data for all AGOs analyzed in this study indicated that the most frequent 5′ nucleotide of tRFs was G [31]. In synchronized *Bright yellow 2* (BY2) cells, a series of 19 nt tRFs were reported to target Gypsy transposon elements. Notably, some of these tRFs showed different associations with AGO1 during the cell cycle, suggesting that their loading was tightly regulated during this process [32]. In another study, a 5′-tRNA-Ala-derived sRNA associated with AGO1 to form a functional RISC, which repressed anti-fungal defense by negatively regulating the expression of *CYP71A13* and biosynthesis of camalexin (a major phytoalexin that confers fungal resistance) [30]. Interestingly, loading competition also occurs across kingdoms. Indeed, sRNAs and their partner AGOs have been at the center of the arms race between viruses and host plants. Using their own RNAi system, plants process viral RNAs into functional AGO-loading siRNAs, which typically target their cognate viral RNAs for degradation [22]. To counteract plant defenses, viruses have evolved strategies to successfully survive in their hosts, among the most well-known of which are VSRs. In another counteractive strategy, some virus-derived sRNAs have been reported to combine with host AGOs to target plant transcripts with critical functions in development or stress response; this has been demonstrated in several plant species, including tobacco [33,34], peach [35], and grape [36]. Another recent example involved the interaction between a prokaryote and a eukaryote: Ren et al. found that rhizobial tRFs could hijack AGO1 proteins of the host legume to promote nodule formation [37].

sRNA-AGO loading is dynamic, not only in terms of sRNA types but also developmental stages and cell types. A recent study used highly synchronizable tobacco BY2 cells to identify differentially accumulated sRNAs during different phases of the cell cycle [32]. A small number of AGO1-bound miRNAs changed in abundance during the cell cycle, and these may have an important role in suppressing the expression of cell cycle-regulated transcripts and disease resistance genes at the mRNA level [32]. Cell division and immune responses triggered by defense genes are highly energy-consuming biological processes that must be balanced during plant development [38]. A possible reason for the repression of defense genes by cell cycle-specific miRNAs is that it guarantees a sufficient energy flux towards cell division. The *Arabidopsis* root tip is a well-established model for the study of cell-type-specific processes, as it consists of five cell layers, each of which expresses a marker gene [39]. Cell-specific expression of *Arabidopsis* AGO1 and subsequent AGO1-IP-sRNA-seq have been used to characterize differences in AGO1-loaded miRNAs among individual cell types [40]. Although no specific miRNAs were loaded across all cell layers or strictly specific to a single layer, some of the miRNA paralogs from the same miRNA family (e.g., miR156 family in the stele) tended to show the strongest loading in specific cell layers [40], providing new insight into the details of AGO protein function and miRNA activity. Thus, through dynamic loading of different sRNAs, AGO proteins can exert highly specific regulatory functions.

### 2.2. Cross-Regulation within the AGO Protein Family

AGO loading not only balances various types of sRNAs but also mediates competition and cooperation within the AGO protein family itself. The predominant sRNA-loading component AGO1 is crucial for sRNA-mediated gene silencing, and its abundance is closely monitored in vivo. To maintain an appropriate AGO1 level, AGO1 is auto-regulated by a negative feedback loop in which miR168 serves as the counterpart miRNA to adjust the AGO1 protein level [41]. AGO1 is one of the main AGOs involved in defense against RNA viruses because it loads virus-derived siRNAs to repress viral RNA. In rice, the grass-specific AGO18 was induced by viral infection and sequestered miR168 to prevent its loading into AGO1, thereby increasing the abundance of AGO1 and promoting its antiviral functions [42]. Likewise, AGO18, which lacks a nuclear localization signal (NLS), has also been speculated to sequester 24 nt phase secondary small interfering RNA (phasiRNA) in the cytosol to affect reproductive development in the grass lineage [8]. Loading similarity between AGO1 and AGO18 was also observed in maize: ZmAGO18 preferred to bind miRNAs and phasiRNAs that are similar in length and 5′ nucleotide composition to those sorted into AGO1 [43]. As a counter-defense, viruses were shown to induce high levels of miR168 in infected zones of *Arabidopsis*; the authors suggested that this inducible miR168 may have been mainly sorted into AGO10, which then acted as a negative regulator of AGO1 through translational repression [44]. Compared with the *ago1* single mutant, the *ago1 ago10* double mutant showed upregulation of AGO1 protein [45].

Cross-regulation between AGO proteins can also provide a backup strategy for plant defense. For example, AGO2 is targeted by miR403 and then loaded into AGO1 [46,47], and it has been hypothesized that AGO2 is activated and plays a redundant antiviral role when AGO1 is disturbed by viral infection [48]. However, other factors may also contribute to the induction of AGO2 during the defense response. In a recent study, chemical pre-treatment was used to trigger systemic acquired resistance (SAR), and the results indicated that AGO2 expression was enhanced during this process through changes in histone modification of the *AGO2* promoter rather than by suppression of AGO1 [49].

AGO1 and AGO10/ZWILLE (ZLL) are two closely related members of the AGO1/5/10 clade. Double mutant analyses have provided genetic evidence that AGO1 and AGO10 function redundantly in embryogenesis and stem cell regulation [50]. They also act redundantly to bind miR172 and, thereby, repress AP2 expression in floral meristems [51]. However, biochemical data have shown that AGO10 and AGO1 have antagonistic functions in terms of regulating miR165/166 activity [52]. miR165/166 are loaded onto AGO1 to suppress *class III homeodomain-leucine zipper* (*HD-ZIP III*) transcription factors, which are essential for stem cell maintenance [53]. AGO10 shows a higher binding affinity than AGO1 for miR165/166 due to instability in the 3′ half of the miR166/miR166* duplex [52], and recent research has demonstrated the structural basis for duplex recognition by a central loop in AGO10 [12]. Surprisingly, the slicer activity of AGO10 is not required for shoot apical meristem (SAM) development [52]. AGO10 appears to be expressed specifically in the provasculature beneath the SAM and acts as a spatial barrier to sequester miR165/166 from loading into AGO1, thus enabling HD-ZIP III function and stem cell maintenance [52,54]. Another study showed that loading of miR165/166 into AGO10 led to subsequent miR165/166 degradation by the exonucleases SDN1 and SDN2 [55].

To resolve the conflicting models of AGO1 and AGO10 function in SAM development, it is important to dissect the mechanism by which *HD-ZIP III* levels are maintained when AGO1 is expressed in the absence of AGO10. Consistent with the functional redundancy model, increased *AGO1* expression driven by the *AGO10* promoter or other domain-specific embryonic promoters rescued the stem cell defects of *zll-1* [26]. Interestingly, *HD-ZIP III* transcripts remained at a high level despite accumulation of miR165/166 in the shoot meristem region of phenotype-rescued transgenic lines [26]. Additional genetic studies demonstrated that uncoupling of AGO1 level from miR165/166 abundance occurred because of the limited supply of RISC chaperones [26]. Thus, to maintain the expression of essential *HD-ZIP III* transcription factors in the SAM during development, the slicing activity of AGO1–miR165/166 is tightly regulated by the sequestration of AGO10 and the limited supply of assembly factors. In addition, miR168–AGO1 auto-repression participates in the control of RISC homeostasis [26].

Tissue-specific expression patterns also reflect the complex loading selection of different AGO proteins. AGO1 and AGO5 were found to incorporate miR156 in different tissues to regulate distinct developmental stages [56]. miR156 has been shown to suppress the vegetative-to-reproductive phase transition [57,58], and AGO1 was the first candidate identified as an effector of miR156 in seedlings [59,60]. Severe pleiotropic developmental defects of the *Arabidopsis ago1* mutant suggest a still-unconfirmed role for AGO1 in miR156-associated regulation of bolting time [56]. AGO5, proposed as a possible angiosperm-specific AGO [7], was recently reported to bind miR156 in the SAM to fine-tune flowering time [56]. Compared with the ubiquitously expressed AGO1, AGO5 is highly expressed specifically in the SAM. *ago5* mutants showed an early flowering phenotype, and an IP assay revealed a physical interaction between AGO5 and miR156 [56]. miR156 overexpression in the *ago5* background abolished the late-flowering phenotype, although changes in leaf architecture caused by excess miR156 were observed. This result indicated that another AGO, probably AGO1, loaded miR156 in leaf tissues and performed a gene-silencing function [56]. Most *ago1* mutants exhibit delayed bolting [61], given the high expression of AGO1 in the SAM, but whether AGO5 has a stronger affinity than AGO1 for miR156 in the SAM still remains to be determined.

The interplay between AGO5 and AGO9 in female reproduction provides an example of two AGO proteins that function antagonistically to regulate gametophyte development. Megasporogenesis and megagametogenesis are two sequential phases of female gamete development. In the first phase, four megaspores are generated by meiosis of a somatic ovule cell, the megaspore mother cell (MMC). Three of the megaspores degrade, and the surviving megaspore undergoes megagametogenesis to produce a multicellular female gametophyte. AGO9 and 24 nt siRNAs work together to prevent sub-epidermal somatic cells around the MMC from adopting megaspore-like identity [62]. Later, AGO5 promotes the initiation of megagametogenesis in the functional megaspore, probably also acting through 24 nt siRNAs [63]. However, AGO5 and AGO9 show different 5′-terminal binding preferences, suggesting that their opposite effects on female reproductive development may be mediated by different classes of siRNAs [63].

Plants can trigger degradation of AGO protein to avoid misloading of unintended sRNAs and adjust the function of the RISC. phasiRNAs are abundant and have been shown to regulate reproductive development in many staple crops [64]. *MEIOSIS ARRESTED AT LEPTOTENE1* (*MEL1*), which encodes the rice paralog of *AGO5*, is germline specific and responsible for phasiRNA binding [65]. MEL1 protein levels are tightly temporally regulated by a monocot-specific E3 ligase that mediates their proteasomal degradation in the late sporogenesis stage [66]. This degradation mechanism ensures accurate cleavage of target genes mediated by MEL1 during the appropriate stage of meiosis, which is critical for male fertility.

### 2.3. sRNA Trafficking

One of the key features of sRNAs is their non-cell autonomous action [67]. In addition to their role as core effectors in classical TGS and PTGS regulation, AGO proteins also participate as sRNA carriers in sRNA movement, both within subcellular compartments and from cell to cell. *MIR* genes are transcribed in the nucleus and processed by the nuclear dicing bodies (D-bodies) to generate mature miRNAs, whereas RISC-mediated gene silencing occurs in the cytosol [68]. For a long time, the mechanism of plant miRNA translocation has been controversial. In metazoans, *EXPORTIN5* (*EXPO5*) mediates the transport of pre-miRNAs from the nucleus to the cytoplasm before the production of mature miRNAs by Dicer [69]. *HASTY* is the plant ortholog of *EXPO5*, and a HASTY-dependent model of miRNA export was initially proposed on the basis of its perinuclear distribution and interaction with RAN (a regulator of karyopherin interactions with cargo molecules) [70]. However, further studies found no alteration in the partition of miRNAs between the nucleus and cytoplasm in a *hasty* mutant [71]. In 2018, Bologna et al. proposed a revised model of nucleo-cytosolic shuttling mediated by cargo-dependent conformational rearrangements of AGO1 [72]. In this model, miRNAs are loaded into AGO1 in the nucleus, exposing the nuclear export signal (NES) in the N-coil region of AGO1 and promoting the export of miRNA-induced silencing complexes (miRISCs) to the cytosol [72]. However, because nuclear membranes serve as a barrier to diffusion of RISCs to the cytoplasm, active nuclear export of these complexes is mediated by the CRM1/EXPORTIN1(EXPO1) pathway [72]. Leptomycin-B treatment, which specifically inhibits the CRM1 pathway, was used to demonstrate that AGO1 assists in miRNA export in a CRM1-dependent manner [72]. Recent work indicates that CRM1 may associate with the nucleoporin NUP1 as well as a core subunit of the TREX-2 complex, which functions in both *MIR* transcription and pre-miRNA processing, to facilitate the nuclear export of miRISCs [73,74].

Heterochromatic siRNAs (hc-siRNAs), 24 nt in length, associate with AGO4 and are responsible for RNA-directed DNA methylation (RdDM). Because both siRNA biogenesis and DNA methylation are nuclear processes, the production and loading of hc-siRNAs were originally thought to be coupled and occur solely in the nucleus [75]. Unexpectedly, nuclear/cytoplasmic isolation and sRNA characterization revealed that hc-siRNA loading into AGO4 occurs in the cytosol, potentially adding a control point prior to the effector stage in the nucleus. In this process, AGO4 was involved in the shuttling of siRNAs back to the nucleus [17]. Likewise, HSP90 was shown to promote siRNA loading into AGO4 in the cytosol [17], changing the AGO4 protein configuration to expose its internal NLS and enable nuclear import. Given the restrictions imposed by the nuclear envelope, whether other factors also promote this process remains to be explored. Nuclear transport receptors such as importins have been proposed to participate in nuclear import of viroid RNA [76], but their roles in AGO4–siRNA nucleo-cytoplasmic transport are unknown.

The short-distance, cell-to-cell movement of miRNAs was initially thought to occur by passive diffusion through the plasmodesmata, producing a gradient distribution pattern [77,78]. However, mathematical modeling excluded the possibility that passive diffusion could produce the miR165/6 activity gradient observed in the root [79]. Another report indicated that miRNA movement was directional and was precisely controlled at defined cell–cell interfaces [80]. P19 is a type of VSR that binds specifically to 21 or 22 nt sRNA duplexes, acting upstream of AGO loading [81]. The 5′ nucleotide composition of siRNAs bound by P19 clearly differed between siRNA incipient cells and siRNA recipient cells, suggesting that the loading of siRNAs into AGOs may hinder their movement to subsequent recipient cell layers [82]. Consistent with this notion, 5′-U siRNA consumption was abolished in the sRNA-loading-deficient *ago1* mutant, suggesting AGO-mediated siRNA removal via loading coupled with siRNA duplex movement [82]. According to this model, elements that coordinate sRNA loading should affect the range of sRNA movement. Indeed, *KATANIN1* (*KTN1*), which regulates microtubule dynamics, also affects miRNA-mediated translational repression [83]. The mechanical separation of source and recipient leaf tissues revealed that the levels of an artificial miRNA were higher in the tissues where it was expressed but lower in the recipient tissues where it diffused in a *ktn1* mutant relative to wild-type plants. To determine how KTN1 regulates miRNA movement, additional experiments were performed on the root system. In the *ktn1* background, miR165/166 loading into AGO1 was enhanced in source cells. A model for cell-to-cell miRNA movement was, therefore, proposed: KTN1 promotes miRNA movement by inhibiting miRNA loading into AGO in the cytoplasm, enabling miRNAs to exit source cells by avoiding cytoplasmic miRISC assembly [84]. Consistent with this model, AGO10 loading sequesters miR398, preventing it from moving out of a specific region of the ovule [85]. Among the AGO10 immunoprecipitation products, miR398 was the second most highly enriched miRNA species [55]. In situ results showed that localization of miR398 was expanded in an *ago10* loss-of-function mutant, whereas miR398 expression was restricted to the chalaza of wild-type ovules [85]. Thus, the spatiotemporal expression and abundance of AGOs add an additional regulatory layer to the control of sRNA mobility.

During the plant reproductive phase, siRNAs are transferred from the vegetative cell to the sperm cell to repress transposable elements, which is a crucial step for epigenetic reprogramming and plant reproductive development. The loss of function of germ-line-specific AGO9 led to a decrease in the levels of ARID1 (a pollen-enriched transcription factor) in the generative cell [86]. In this work, researchers developed a reporter system in which the successful translocation of siRNAs from the vegetative nuclei to the sperm silenced the expression of a GFP reporter gene in the sperm line. With this system, researchers could monitor siRNA translocation efficiency by measuring GFP protein level. They found that both AGO9 and ARID1 reinforced heterochromatic silencing by facilitating siRNA mobility [86]. Future biochemical studies will be necessary to elucidate the mechanism by which AGO9 participates in this movement process.

Trans-kingdom RNA interference (RNAi) is mediated by the movement of sRNAs between different species and has been used as a biological control strategy for pests and pathogens [87]. Host-induced gene silencing (HIGS) is developed through transgenic manipulation, in which pathogen-derived, double-stranded RNA fragments are transcribed and processed to siRNAs in host plants, then transported into pathogens to target crucial metabolic genes (e.g., lethal genes). Nonetheless, the mechanism of interspecies sRNA movement between plants and pathogen cells is largely unknown. Extracellular vesicles (EVs) are small membrane-bound compartments produced by both mammalian and plant cells; they are released into the extracellular matrix carrying cargoes that include metabolites, proteins, lipids, and RNAs as intercellular communication signals [88]. EVs, thus, provide an excellent means for interspecies shuttling of sRNA while avoiding degradation by extracellular RNases. sRNAs have been reported to localize inside EVs, which mediate their transport between plants and interacting organisms [89]. Mass spectrometry analysis of isolated EVs from fungus-infected *Arabidopsis* leaves detected a number of RNA binding proteins, such as AGO1 and RNA HELICASE11/37, which may be responsible for the selective loading of sRNAs into plant EVs [89]. Levels of EV-enriched sRNAs, rather than total sRNAs, were lower in *ago1* and other RNA-binding-protein mutants than in wild-type plants, indicating that their transport into the fungal pathogen was less efficient [89].

However, other researchers found that sRNAs, including some kinds of siRNAs, were present in apoplastic wash fluid in the form of RNA–protein complexes rather than packaged inside EVs [90,91]. Thus, it remains controversial whether sRNAs are located inside EVs or are present only as contaminants that are adhered to the outer EV surface. Because the association of extracellular RNAs with proteins could protect them from degradation in the complex extracellular environment, protease treatment combined with RNase digestion [92], which will not disrupt EVs, may help to distinguish the RNA–protein complexes present inside and outside EVs. With this method, Zand Karimi et al. found that most miRNA and siRNA species were located outside EVs in the apoplast, protected by their binding proteins [93]. Interestingly, AGO2 and the putative m^6^A-binding protein GRP7 were co-immunoprecipitated with extracellular lncRNAs in this study. Consistent with their roles in the secretion and stabilization of extracellular RNAs, apoplastic lncRNAs and sRNAs were significantly less abundant in AGO2 and GRP7 mutants [93]. There is a pressing need for the development and evaluation of uniform experimental standards to enable further research on interspecies sRNA transfer and extracellular movement.

In general, AGO1 is thought to act in the cell autonomously, as demonstrated in root tips with well-defined, cell-type-specific markers. Fluorescent signals indicated that the expression of GFP-tagged AGO1 driven by a cell-layer-specific promoter was restricted to the corresponding cell layer [40]. Intriguingly, two independent groups recently reported that not only sRNA cargoes but also the AGO protein itself could traffic between adjacent cells or different cell types, as demonstrated for two AGO1 sub-clade members. Comparing with the positions of their mRNA transcripts and mature proteins revealed that AGO1b and AGO1d serve as mobile signals during rice reproductive development [94,95]. These new findings suggest that there may be distinct properties of AGO proteins in different plant tissues. But what are the detailed mechanism(s)? Does the movement of AGO proteins require other auxiliary factors? Resolution of these questions will shed light on the non-cell-autonomous activity of AGOs and sRNAs during anther development.

### 2.4. New Roles

Although the biological functions of AGOs have been intensively studied in past decades, because of the diversity of this large family and the independent expansion of each subclade after species differentiation [9], the overall functions of AGO proteins are still underestimated. AGO8 was long thought to be a pseudogene in *Arabidopsis* because of a predicted splice-inducing frame shift in its putative coding sequence, which suggested it would form a truncated and nonfunctional protein [96]. However, cytological analysis revealed a more frequent occurrence of mutant ovules in two independent *ago8* mutant alleles, and other mutants in the AGO4 clade, including *ago4*, *ago6*, and *ago9*, showed a similar phenotype. Quantitative real-time PCR revealed upregulation of *AGO8* in ovules of *ago4 ago9*, implying that AGO8 might play a compensatory role during female gametophyte development [97]. Assays with fluorescent-protein-tagged AGO8 revealed that it was specifically expressed in egg cells of *Arabidopsis* and was localized in the cytoplasm [98]. Although AGOs in the same clade typically show similar functions, AGO3 from the AGO2/3/7 clade (the last characterized AGO member in *Arabidopsis*) is an exception. The activity of AGO3 is similar to that of AGO4, and it binds 24 nt siRNAs that overlap with those of AGO4, but it differs in function from its closest paralog, AGO2, which functions mainly in antiviral defense [99]. Unexpected functions of AGO8 in tobacco defense against herbivory have also been revealed. AGO8-silenced transgenic *Nicotiana attenuata* are hyper-susceptible to herbivore attack, owing to reduced biosynthesis of defense metabolites. Although the differential expression of miRNAs and target genes in response to AGO8 silencing as well as the protein structural basis for sRNA loading into AGO8 were analyzed, evidence for direct AGO8 protein–sRNA interaction remains to be generated [100].

The nuclear roles of AGO proteins, including roles in DNA repair and transcription, have recently been summarized in a review by Bajczyk [101]. Of particular interest are new findings suggesting that AGO proteins may play transcription-factor-like roles. Several plant AGO proteins can directly bind to chromatin to regulate gene expression. A chromatin immunoprecipitation (ChIP) assay of AGO1 showed a chromatin occupancy pattern analogous to that of RNA polymerase II (Pol II), suggesting that AGO1 can promote gene transcription, especially in response to plant hormones and stresses [102]. To date, the molecular mechanisms by which AGO1 helps to regulate the transcription of nuclear genes remain unclear. The previous report proposed that the chromatin remodeling factor SWI2/SNF2 and DCL1-dependent sRNAs, which are probably not miRNAs, may participate in this process [102]. However, a later study found that the ATPase subunit of SWI2/SNF2 was involved in the production of miRNAs [103]. Generation of the sRNAs required for AGO1 chromatin binding appeared to be dependent on the miRNA biogenesis machinery [102]. Thus, impaired chromatin binding of AGO1 in mutants of SWI2/SNF2 complex subunits may be due to misregulation of sRNA biogenesis. A later study in rice revealed a new mechanism of transcriptional modulation mediated by AGO2 [104]. OsAGO2 binds directly to the promoter of *OsHXK1* to regulate ROS generation during anther development by a mechanism that involves DNA methylation [104]. In contrast to AGO1, whether the nuclear regulatory process of AGO2 depends on its associated sRNAs is still unknown. AGO1 also participates in a co-transcriptional repression mechanism that determines plant flowering via the floral repressor locus *FLOWERING LOCUS C* (*FLC*) [105]. AGO1 physically associates with co-transcriptional regulators, such as components of Pol II and splicing-related proteins, to regulate antisense RNA *COOLAIR* processing, ultimately creating a local chromatin-silencing environment that determines the output of the sense transcript, *FLC* [105].

Consistent with its involvement in RNA processing, AGO1 also participates in mRNA intron splicing, a function that is conserved in animals [106]. The binding regions of AGO proteins are enriched at intronic regions of mRNA transcripts, a phenomenon consistent with the intron retention observed in *ago* mutants and the proposed function of AGOs in alternative splicing [107]. IP-mass spectra also revealed several novel mRNA-binding interactors of AGOs that facilitate their intron recognition [107]. Nonetheless, the detailed nuclear regulatory mechanism of AGOs requires further investigation. In human cell cultures, AGO2 participates in mRNA quality control by sensing abnormal and nascent peptide chains in the cytosol [108]; whether this mechanism is conserved in plants remains unclear.

Phase separation is a process in which multiple biomolecules assemble autonomously to form membraneless cellular structures; these independent and flexible sub-environments separate the vast diversity of biochemical reactions that occur within the limited space of a single cell. Intracellular liquid–liquid phase separation (LLPS) occurs during ongoing processing of miRNAs in the D-body [109]. Interestingly, although AGO4 does not contain domains involved in LLPS, it often colocalizes with discrete nuclear speckles called Cajal bodies (important functional nuclear foci for RdDM) [110] and AB-bodies [111]. These two distinct nuclear AGO4 bodies were representatives of phase separation and were structurally independent of one another [111]. Several other AGO proteins (AGO1, AGO2, and AGO3) are predicted to harbor prion-like domains [112] that may provide the driving force for LLPS formation [109]. Although experimental confirmation for a role of AGOs in phase separation is necessary, a study on AGO1 subcellular localization during different phases of the cell cycle revealed a complex distribution pattern with foci in the nucleus and cytosol that is consistent with the LLPS potential of AGO1 [32].

## 3. Perspectives

Considerable research efforts in angiosperms have expanded our understanding of the biological significance of AGOs beyond canonical RNA loading (Figure 1, Table 1), but many outstanding issues remain to be clarified. siRNAs usually originate from the cleavage of long, double-stranded RNAs by Dicer-like (DCL) endoribonucleases to generate mature siRNAs, 21~24 nt in length. However, other than the studies on the 5′ nucleotide preference for siRNA sorting into AGOs and the influence of siRNA-generating loci [13,113], there have been relatively few studies of the mechanism(s) that regulate siRNA loading compared with miRNA loading. Although a very recent study suggested that two conserved domains confer different sRNA selectivity to AGO4 clade members [114], additional factors need to be explored in future work. In particular, the details of the loading selection of AGOs with functionally important siRNAs, such as phasiRNAs, viral-derived siRNAs, tRFs, and even risiRNAs, should be investigated.

A role for AGO proteins in cell-to-cell movement of sRNAs is emerging, but the details remain obscure. A consensus view is that AGOs and other RNA-binding proteins have a protective effect on sRNAs in extracellular environments. Given the broad interest in the role of AGOs in cross-species sRNA movement, a standard workflow to separate EVs will help to answer a current controversial question: are sRNAs secreted from cells in the form of EVs or just protein–RNA complexes? In addition, because the high molecular weight of AGO proteins typically restricts their mobility between cells, what is the mechanism by which AGOs move across different cell types?

The expansion of individual AGO subclades occurred after the formation of various plant lineages, suggesting that there may not be a one-to-one correspondence between phylogenetic position and function [7]. Therefore, it is difficult to predict the roles of AGO proteins from one species on the basis of characterized orthologs in species from other lineages. Research on AGO functions in many unexplored, economically important plants will provide new clues for agricultural improvement. It will also be important to characterize the sRNA species loaded by the AGO-like proteins, which have been lost in angiosperms. Because miRNAs show significant divergence among plant species, it is possible that some novel miRNAs in early terrestrial plants coordinated with this specific AGO-like subclade to perform important functions, such as enabling adaption to the early land environment or promoting the success of early land plants.

## Figures and Tables

**Figure 1 ijms-24-16054-f001:**
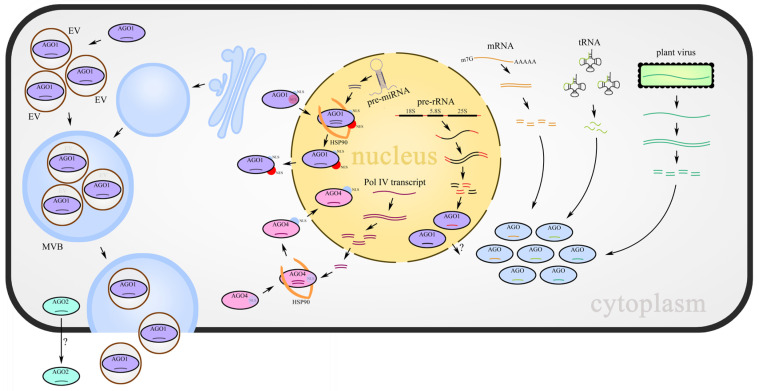
sRNA-AGOs loading provides a balance between distinct classes of sRNAs and participates in small RNA movement. miRNAs produced from MIR genes, risiRNAs derived from aberrant pre-rRNAs, siRNAs derived from abnormal mRNA fragments, tRNA-derived RNA fragments (either from plants or their interacting organisms), and virus-derived sRNAs are loaded into their partner AGOs, respectively or competitively. Cell-to-cell movement of sRNAs is mediated by the AGO2–sRNA complex directly or packaged into EVs with the assistance of AGO1. Loading of miRNAs and 24 nt Pol IV siRNAs cause the configuration change in AGO proteins to expose NES or NLS, which results in sRNAs shuttling between the nucleus and cytoplasm.

**Table 1 ijms-24-16054-t001:** Multiple roles of ARGONAUTE proteins in plants.

Roles	Involved Small RNAs	Involved Plant AGOs	Functions
Loading competition balances sRNA types	miRNAs, siRNAs from PTGS	AGO1	PTGS efficiency control [21]
Loading competition balances sRNA types	aberrant siRNAs	AGO1, AGO2	for degradation of abnormal RNAs by PTGS [27,28]
Loading competition balances sRNA types	plant tRFs	AGO1	repress gene expression and biogenesis of phytoalexin to negatively regulate plant response to biotic stress [30]
Loading competition balances sRNA types	plant tRFs	AGO1	maintain genome stability through targeting TE transcripts, regulating cellular processes such as cell cycle [32]
Loading competition balances sRNA types	rhizobial tRFs	AGO1	regulate plant nodulation by hijacking host AGO1 [37]
Loading competition balances sRNA types	virus-derived sRNAs	AGO1	target plant transcripts with critical functions in development or stress response [33,34,35,36]
Cross-regulation within the AGO protein family	miR168	AGO1, AGO18	virus counter-defense plant immunity [42]
Cross-regulation within the AGO protein family	miR403	AGO1, AGO2	a backup strategy for plant defense [48]
Cross-regulation within the AGO protein family	miR165/166	AGO1, AGO10	provide spatial barrier for shoot apical meristem (SAM) development regulation [52,54]
Cross-regulation within the AGO protein family	miR156	AGO1, AGO5	Tissue-specific selection of miRNA by AGOs to regulate flowering [56]
Cross-regulation within the AGO protein family	24 nt siRNAs	AGO5, AGO9	female gametophyte development [63]
sRNA trafficking	miRNAs	AGO1	nucleo-cytosolic shuttling of miRNAs [72]
sRNA trafficking	hc-siRNAs	AGO4	nucleo-cytosolic shuttling of siRNAs [17]
sRNA trafficking	miRNAs	AGO1, AGO2/4, AGO5	short-distance cell-to-cell movement of both miRNAs and siRNAs [82,84]
sRNA trafficking	miR398	AGO10	miR398 distribution region was limited in a specific site of ovule [85]
sRNA trafficking	siRNAs derived from vegetative cells	AGO9	siRNAs moved into the sperm cells to repress transposable elements [86]
sRNA trafficking	miRNAs, siRNAs	AGO1	be responsible for selective loading of sRNAs into plant EVs [89]
sRNA trafficking	miRNAs, siRNAs, lncRNAs	AGO2	the secretion and stabilization of extracellular RNAs without packaging into EVs [93]
New roles	unknown	AGO8	female gametophyte development, tobacco defense against herbivory [97,100]
New roles	24 nt siRNAs	AGO3	regulate epigenetic silencing by DNA methylation [99]
New roles	nuclear AGO1-associated sRNAs	AGO1	binding directly to the transcription start sites to promoter gene transcription [102]
New roles	unknown	AGO1, AGO2	associated with promoter or co-transcriptional regulators to modulate mRNA/antisense RNA transcription [104,105]
New roles	probably miRNAs	AGO1	mRNA intron splicing [107]
New roles	miRNAs, siRNAs	AGO1, AGO4	Phase separation, processing of miRNAs in the D-body and RdDM [109,112]

## Data Availability

Not applicable.

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
