# Peer review of "Beyond Loading: Functions of Plant ARGONAUTE Proteins"

_ijms, 2023, doi:10.3390/ijms242216054_

Round 1
Reviewer 1 Report
Comments and Suggestions for Authors
The aim of the review manuscript entitled “Beyond Loading: Functions of Plant ARGONAUTE Proteins”, is to provide a summary of the recent research progress on the nontraditional functions of AGO proteins in plants. The authors provide a summary of the different sRNA pathways, cross-regulation of different RISC activities during plant development and defense, and the roles of AGO proteins in sRNA movement.
Argonaute proteins play a pivotal role in RNA silencing mechanisms, a fundamental process in plants. This system regulates gene expression and defends against viral infections and transposon activity. In plants, Argonaute proteins are integral components of small RNA-mediated pathways, including microRNAs (miRNAs) and small interfering RNAs (siRNAs). Argonaute proteins function by binding to small RNA molecules, guiding them to complementary target RNA sequences. This binding induces gene silencing through either mRNA degradation or translational repression. In plants, miRNAs are generated from precursor transcripts through a series of enzymatic processes. These mature miRNAs are then loaded onto Argonaute proteins, enabling them to recognize and silence target mRNAs. Argonautes also play a crucial role in plant development, growth, and stress responses. They regulate various processes, including leaf polarity, flower development, and root growth. Additionally, they are vital in responses to biotic and abiotic stresses, such as pathogen defense and drought tolerance.
While many reviews have focused on the traditional roles of AGO in RNA silencing, the major strength of this review is that it provides a summary of the recent emerging roles of AGO in plants. The manuscript is well-written, and while it is not a comprehensive review of the additional roles of AGO in plants, it offers a concise summary of what is currently known. It is meticulously crafted and thoughtfully articulated; it navigates the intricate terrain of its subject matter with precision and finesse. The author's command over the topic is evident from the outset, as they deftly weave together the research in the area, offering insightful perspectives and nuanced analyses. With a judicious selection of references and a seamless flow, this review serves as an indispensable resource for those delving into the field.
However, there are a few points that if incorporated will strengthen this review.
* In Section 2.1, Loading competition balances sRNA types, the authors should elaborate on the actual mechanism of sRNA loading on Argonaute proteins. Even though it is beyond the scope of this review, a summary of the loading mechanism will be beneficial.
· The authors should consider providing a section in which they discuss more comprehensively the role of virus suppressor proteins in silencing mechanisms.
· The authors should consider creating a summary in the format of a table in which they summarize the Argonaute proteins, their known traditional roles, and their emerging non-traditional roles.
Reviewer 2 Report
Comments and Suggestions for Authors
The manuscript under review provides an insightful overview of the multifaceted roles of ARGONAUTE (AGO) proteins in the RNA-induced silencing complex (RISC), shedding light on both canonical and non-canonical functions. The paperwork highlights AGO proteins' central role in gene silencing by loading small-RNA (sRNA) cargoes. This introductory statement effectively sets the stage for readers, emphasizing the significance of AGOs in the context of RNA interference. The authors bring a fresh perspective to the review by emphasizing traditional roles and the nuanced aspects of AGOs. They discuss the balance among different sRNA pathways, the cross-regulation of RISC activities during plant development and defence, and the emerging roles of AGOs in sRNA movement. This broad view is commendable as it adds depth to our understanding of AGO functions in diverse biological contexts.
Moreover, mentioning non-canonical functions of plant AGOs introduces an element of novelty. The paperwork hints at the evolving nature of our understanding of AGO proteins, suggesting that these proteins may have even more undiscovered roles in cellular processes. In conclusion, this review article appears promising for researchers and scientists interested in the versatile functions of AGO proteins. It offers a comprehensive overview of established and emerging roles, serving as an excellent resource for future studies in this field. The abstract effectively conveys the critical points of the review, inviting readers to explore the full article for a more detailed understanding of ARGONAUTE proteins' significance in eukaryotic gene regulation.
Comments on the Quality of English LanguageEven if the article is difficult to follow due to multiple abbreviations, the English language is used correctly with no detectable editing errors.
